

# Irradiation with carbon ion beams affects soybean nutritional quality in early generations

Changkai Liu[1], Xue Wang[1,2], Yansheng Li[1], Heng Chen[1,2], Qiuying Zhang[1,3] and Xiaobing Liu[1]

[1] Key Laboratory of Mollisols Agroecology, Northeast Institute of Geography and Agroecology, CAS, Harbin, China
[2] University of Chinese Academy of Sciences, Beijing, China
[3] Innovation Academy for Seed Design, CAS, Harbin, China

## ABSTRACT

As people's demand for healthy diet increases, improving soybean seed nutritional quality is becoming as important as yield. Carbon ion beam radiation (CIBR) is an effective method to create soybean mutants, and thus breeding cultivars with better seed nutritional quality. In this study, the high-yield soybean line 'Dongsheng 28' was used, and three CIBR doses (100, 120, and 140 Gy) were used to explore the characteristics of quality separation and variation in the offspring of early mutant populations. Eleven quality traits, including protein, oil, sucrose, soluble sugar, iron (Fe), manganese (Mn), zinc (Zn), cupper (Cu), daidzin, glycitin, and genistin concentrations were analyzed in the $M_2$ and $M_3$ generations. The results revealed that the range of protein and oil concentration of all three CIBR doses changed by 38.5–42.9% and 18.8–23.8% in the $M_2$ and $M_3$ generations, respectively, while soluble sugar and sucrose concentrations changed by 48.1–123.4 and 22.7–74.7 mg/g, with significant effects by 140 Gy across the two generations. Therefore, around the optimum range, a higher CIBR dose is better for high protein, oil, and sugar varieties selection. In general, irradiation raised isoflavone concentrations, but 140 Gy had an inhibitory effect on isoflavone concentrations in the $M_3$ generation. Although a variety could not be released in the $M_2$ or $M_3$ generation, the results of this study have important guiding significance for the targeted cultivation of specific nutritional quality materials. For instance, a lower irradiation dose is preferable when breeding targets are higher isoflavones and Mn concentrations. It is essential to increase the irradiation dose if the breeding targets contain high levels of protein, oil, sucrose, soluble sugars, Fe, Zn, and Cu.

Corresponding author
Qiuying Zhang,
zhangqiuying@iga.ac.cn

# INTRODUCTION

Soybean is a unique legume crop because of its diverse nutritional value including protein, oil, isoflavones, trace elements, and metabolizable energy (*Schmutz et al., 2010*; *Liu et al., 2019*). The seed protein concentration of soybean is about 40%, which is about 2–4 times higher than that of corn, rice and wheat. Soybean seed oil concentration is about 20%, and contains eight kinds of amino acids necessary for human, especially lysine and tryptophan

which cannot be synthesized by human body (*Patil et al., 2017*). Soybean seeds contain about 20–30% carbohydrates, and their complex composition provides many functions (*Cober & Voldeng, 2000*). The ratio of sucrose, starch and dietary fiber in soybean influences its nutritional value greatly (*Karr-Lilienthal et al., 2005*). Soybean seeds are also rich in isoflavones and trace elements, which are essential for the human diet (*Wu et al., 2020*).

Crop breeding relies heavily on plant germplasm. Mutation breeding has a unique position in breaking the bottleneck of germplasm resources, thus creating more beneficial resources than cross breeding. Higher mutation rate and wider mutation spectrum of soybean seed nutritional quality have been reported by physical mutagenesis and chemical mutagenesis (*Espina et al., 2018*). With higher linear energy transfer (LET), higher mutation rates, and wider mutation spectra under lighter damage, carbon ion beams (CIBs) are becoming increasingly popular for mutation breeding in a variety of plants, including soybean (*Arase et al., 2011*). It could therefore be easier to breed high-quality soybean varieties with the help of studying the effects of carbon ion beam irradiation on soybean seed nutritional quality traits.

There was evidence that combined γ-radiation and ethylmethanesulfonate (EMS) improved the concentration of soybean oil with higher levels of oleic acid but lower levels of linolenic acid (*Patil et al., 2007*). It has also been reported that irradiation of grain cereals and leguminous crops seeds leads to increased protein content and higher carbohydrate and vitamin levels (*Jan et al., 2012*). *Mikuriya et al. (2017)* demonstrated that the populations mutagenized by carbon ion beam irradiation exhibited lower isoflavone concentration in soybean seeds, which was closely related to the reduction of leaf chlorophyll concentration. The increased concentration of genistin, genistein, daidzin and glycitein, but decreased concentration of glycitein and daidzein in soybean seedlings were found by laser irradiation (*Jin et al., 2011*). In soybean seedlings, Fe, Cu, and Zn concentrations were increased with the increased irradiation doses (*Alikamanoglu, Yaycili & Sen, 2011*). Thus, carbon ion beam irradiation may have greater potential for accelerating high-quality soybean breeding. However, knowledge on soybean populations obtained from CIBs is still lack, especially concerning seed trace elements and isoflavones.

Previous studies have identified that 120 Gy was the optimal irradiation dose for Dongsheng 28 due to its relatively appropriate mortality (*Wang et al., 2021a*). Therefore, around the optimal irradiation dose, increasing or decreasing the dose might have specific effects on the selection of soybean quality traits. The present study investigated the seed protein, oil, isoflavones, carbohydrates and trace elements in $M_2$ and $M_3$ generation obtained by carbon ion beam mutagen treatment, the aim was to develop soybean lines with beneficially altered seed composition.

## MATERIALS AND METHODS

### Plant material and experiment design

The experimental material was variety 'Dongsheng 28', which was bred by the Northeast Institute of Geography and Agroecology, Chinese Academy of Sciences. The average plant height of Dongsheng 28 was about 100 cm, with no branching and semi-indeterminate

growth type. Dongsheng 28 had a yellow seed coat, 18 g of 100-seed weight, and 125-day growth cycles.

In 2018, the irradiation was directly targeted at the hilum of each seed with 960 Mev carbon ion beam. The mutagenesis dose was 0 (control), 100, 120 and 140 Gy respectively and was carried out at the Institute of Modern Physics, Chinese Academy of Sciences. Referring to the practice of *Williams & Hanway (1961)*, *Arase et al. (2011)*, and *Mikuriya et al. (2017)*, one hundred seeds were treated at each radiation dose. After that, the first generation ($M_1$) was planted in May 2018 at the Agronomy Farm of the Northeast Institute of Geography and Agroecology, Chinese Academy of Science (45°73′N, 126°61′E). The seeds of all survival plants were harvested separately in September. In total, 47 plants in the treatment of the 100 Gy group, 26 plants in the treatment of the 120 Gy group and 18 plants in the treatment of the 140 Gy group were collected. In May 2019, the seeds of individual plants harvested from the last generation were sown in single lines ($M_2$ generation), ranging from 55 to 333 seeds for each line. At harvest, two pods were harvested from every individual plant to form a block with three replicates for each single line in September (*Wang et al., 2021a*). Half of the blocks were used for the determination of seed nutritional quality and the other half were saved as seed for the next generation. In May 2020, the $M_3$ generation was also sown in single lines, 100 seeds of each line were randomly selected, 47 lines for 100 Gy, 26 lines for 120 Gy, and 18 lines for 140 Gy treatments, respectively. The sample collection method for the $M_3$ generation was the same as the $M_2$ generation.

The planted field was a typical black soil with 29.3 g kg$^{-1}$ organic matter, 2.4 g kg$^{-1}$ total N, 1.5 g kg$^{-1}$ total P and 18.8 g kg$^{-1}$ total K. Before seeding, 70 kg ha$^{-1}$ diammonium phosphate, 98 kg ha$^{-1}$ urea and 120 kg ha$^{-1}$ potassium sulfate base fertilizers were applied. The seeds were sown in a row with 45 cm spacing and 5 cm plant spacing. The local normal management for weed control and other agronomic practices were adopted in the experiment.

## Chemical analysis of samples

### Crude protein

The seed crude protein concentration was determined using the method of combustion nitrogen analysis by Elementar-Vario (Elementar Analysensysteme GmbH E-III, Germany) (*Li et al., 2012*). A conversion factor of 6.25 was used to convert total nitrogen to crude protein concentration (*Saldivar et al., 2011*). Crude protein concentration = 6.25 × total nitrogen concentration.

### Crude oil

Total oil concentration in seeds was determined by the Soxhlet extraction method. Approximately 0.5 g of dried soybean seed sample with a piece of weighed filter paper was wrapped up, then put in a Soxhlet apparatus in a 60 °C waterbath, and adding 200 mL ethyl ether to the Soxhlet apparatus for extracting oil. After a 48-h extraction, the defatted sample was placed in an oven at 45 °C about 12 h. The crude oil concentration was calculated by the difference method according to *Li et al. (2014)*.

### Soluble total sugar and sucrose

The determination of soluble sugar and sucrose was based on the method of *Tu et al. (2017)*. About 0.5 g sample was extracted by 4 mL 80% ethanol, and placed in a 80 °C water bath for 30 min. Then the mixture was centrifuged at 4,500 r/min$^{-1}$ for 3 min, and the supernatant was removed to a new 15 mL -centrifuge tube. A total of 3 mL 80% ethanol was added to the precipitate and repeated as above operation twice. The supernatant was brought up to a 10 mL final volume. Then 4 mL anthrone (1,000 mL 80% $H_2SO_4$ + 2.5 g anthrone) was added to 1 mL supernatant, and placed in a 90 °C waterbath for 10 min, and measured at 620 nm (Xinshiji T6, Beijing, China). The determination method of sucrose was basically the same as soluble sugar, but before adding anthrone, it was necessary to add 25 μL 12 mol/L of NaOH and put into waterbath at 100 °C for 10 min in order to remove the monosaccharides.

### Trace element

Trace element was determined with the modified method of *Xue et al. (2006)*. About 0.5 g sample was weighed and placed in a crucible. After carbonization, it was placed in a muffle furnace at 510 °C for 2 h. After cooling, 1 mL hydrochloric acid and 1 mL deionized water were added to dissolve the ash. The volume was brought up to 25 mL after filtering. The analysis of the trace element was performed by Atomic Absorption Spectrometry (AAS) (TAS-990, Beijing, China).

### Isoflavones

Isoflavones was determined according to the method described by *Hoeck et al. (2000)* with high-performance liquid chromatography (HPLC). Weighing about 0.5 g sample into a 15 mL centrifuge tube, and adding 9 mL 80% methanol, he sample was then ultrasonically extracted at 60 °C for 30 min, centrifuged at 5,000 rpm for 5 min, and the supernatant was collected into a 25 mL volumetric flask. Adding 6 mL 90% methanol to the precipitation twice. The supernatant was finally diluted to 25 mL with 10% methanol. The filtrate through a 0.45 μm filter membrane was used for the determination of isoflavones. The conditions of HPLC were: RPC18 stainless steel chromatography column; 0.1% acetic acid and 0.1% acetic acid acetonitrile of the mobile phase; 1.0 mL/min of the flow rate; 40 °C of the column temperature and 260 nm of the wavelength.

## Statistical analysis

Excel 2016 and SPSS 25.0 were used for the analysis of statistical data. The frequency distribution histograms were created by SPSS 25.0. A hierarchical clustering method was used to cluster $M_3$ generation mutant lines and a square Euclidean distance was used to measure similarity. The figures of the effects of different irradiation doses on the quality traits were created with Graphpad Prism 8. The line was the average under different irradiation dose. Pearson method was used to analyse the correlation of different nutrition quality traits.

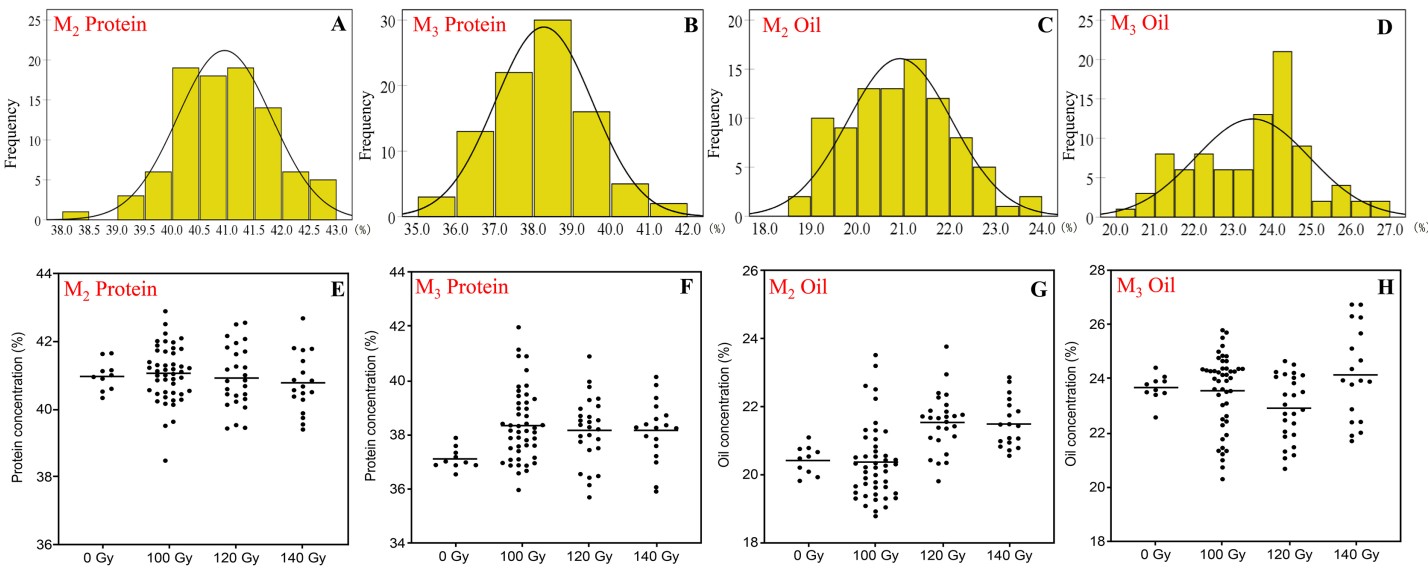

**Figure 1** The distribution of seed protein and oil concentration in $M_2$ and $M_3$ generations. (A and B) Frequency distribution of protein concentration; (C and D) frequency distribution of oil concentration; (E and F) effect of different irradiation doses on protein concentration; (G and H) effect of different irradiation doses on oil concentration; $M_2$ generation: A, C, E and G; $M_3$ generation: B, D, F and H.

# RESULT

## Effects of CIBR on soybean protein and oil concentration

As a result of mutagenic treatments, a wide range of variability for protein concentration and oil concentration was found in the $M_2$ and $M_3$ generation (Fig. 1). The coefficient of variation (CV) of protein and oil was significantly higher than that of control treatment.

In the $M_2$ generation, the range of protein and oil concentration in the three CIBR doses was 38.5–42.9% and 18.8–23.8%, respectively. In this generation, the median of protein concentration in the three CIBR doses was similar with control treatment, while the median of oil concentration exhibited an increased trend in the 120 and 140 Gy groups. The CV of protein ranged 2.0–2.2% in the $M_2$ generation but ranged 5.1–6.9% in the $M_3$ generation. Turn to the $M_3$ generation, seed protein generally increased compared with control. The dispersion degree of oil in each CIBR population increased, and the median of oil in the 120 Gy group decreased significantly.

In the $M_3$ generation, the 140 Gy treatment improved both protein and oil concentrations, and their concentrations were higher in most lines compared to control.

## Effects of CIBR on seed soluble sugar and sucrose concentration

Compared to protein and oil concentration, the variation of the soluble sugar and sucrose concentration was more diverse. Besides, the variations were relatively consistent in the $M_2$ and $M_3$ generations, which was significantly different from that of control treatment.

In terms of the concentration of soluble sugar, the range in the $M_2$ generation and $M_3$ generation was 78.0–123.4 and 48.1–98.8 mg/g, respectively. Compared with the control treatment, mutagenic treatments generally increased the concentration of soluble sugar,

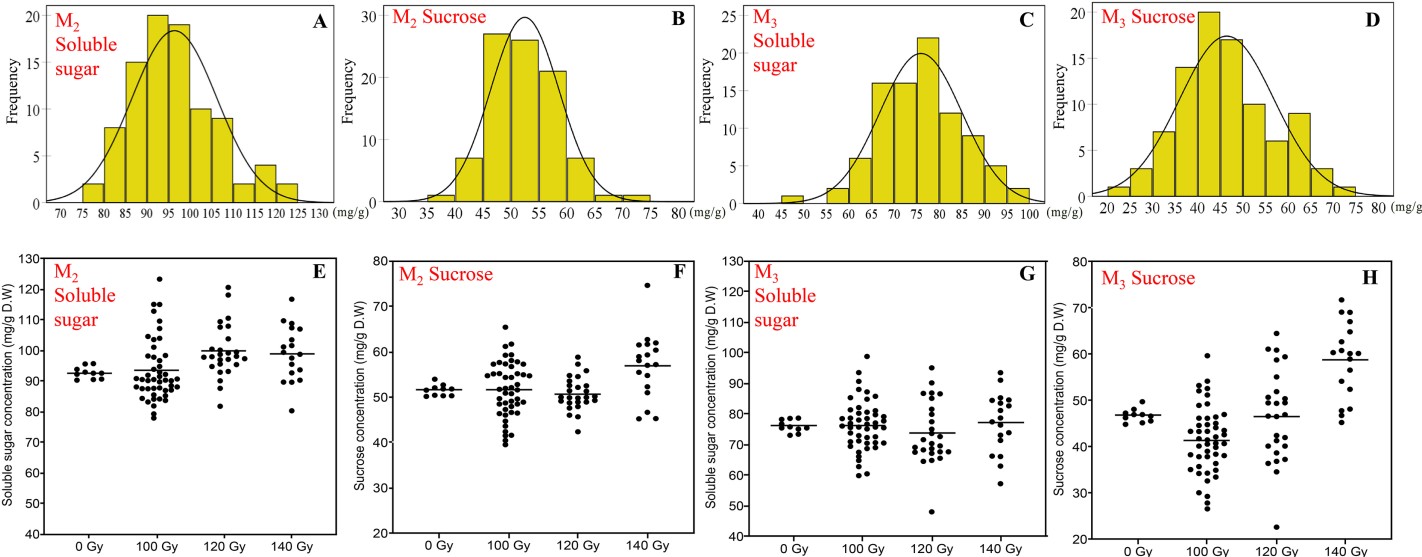

**Figure 2 The distribution of seed soluble sugar and sucrose concentration in M₂ and M₃ generations.** (A and B) Frequency distribution of soluble sugar concentration; (C and D) frequency distribution of sucrose concentration; (E and F) effect of different irradiation doses on soluble sugar concentration; (G and H) effect of different irradiation doses on sucrose concentration; M₂ generation: A, C, E and G; M₃ generation: B, D, F and H.

especially in the 140 Gy treatment. The range of sucrose concentration was 39.6–74.7 and 22.7–71.8 mg/g in the M₂ generation and M₃ generation, respectively. Overall, the distribution of sucrose was basically the same as soluble sugar, higher doses induced a more significant positive effect on sucrose concentration (Fig. 2).

## Effects of CIBR on seed trace element concentrations

Significant separation was found in the changes of Fe, Mn, Zn and Cu concentrations by carbon ion beam irradiation compared with the control (Fig. 3).

In the M₂ generation, Zn concentration ranged 24.9–35.8 µg/g for all CIBR doses. The maximum concentration of Zn by CIBR treatments was 28.7% higher than that of control (27.8 ± 0.53 µg/g). In the M₃ generation, the Zn concentration ranged 31.5 to 42.6 µg/g, and 89.0% mutagenic lines were concentrated at 36.0–42.0 µg/g. In the M₂ generation, compared with control, irradiation treatments showed obvious positive effects on Zn concentration, especially at 100 and 140 Gy doses with the highest CV of 7.25% at 100 Gy. In the M₃ generation, however, higher irradiation dose (120 and 140 Gy) increased Zn concentration.

In the M₂ generation, the range of Fe concentration was 63.7–89.5 µg/g for all CIBR doses. Seed Fe concentration exhibited a trend of 100 < 120 < 140 Gy. In the group of 140 Gy, the median of seed Fe concentration was close to the non-irradiated control (80.66 ± 1.35 µg/g). In the M₃ generation, Fe concentration was different from that of the M₂ generation, and was increased by all CIBR doses. Especially in the 140 Gy group, the Fe concentration of 90% plants was increased compared with the control group.

The range of Mn concentration for all CIBR doses in the M₂ and M₃ generations was 17.1–23.3 and 13.8–32.5 µg/g, respectively. There were fewer differences between the CIBR

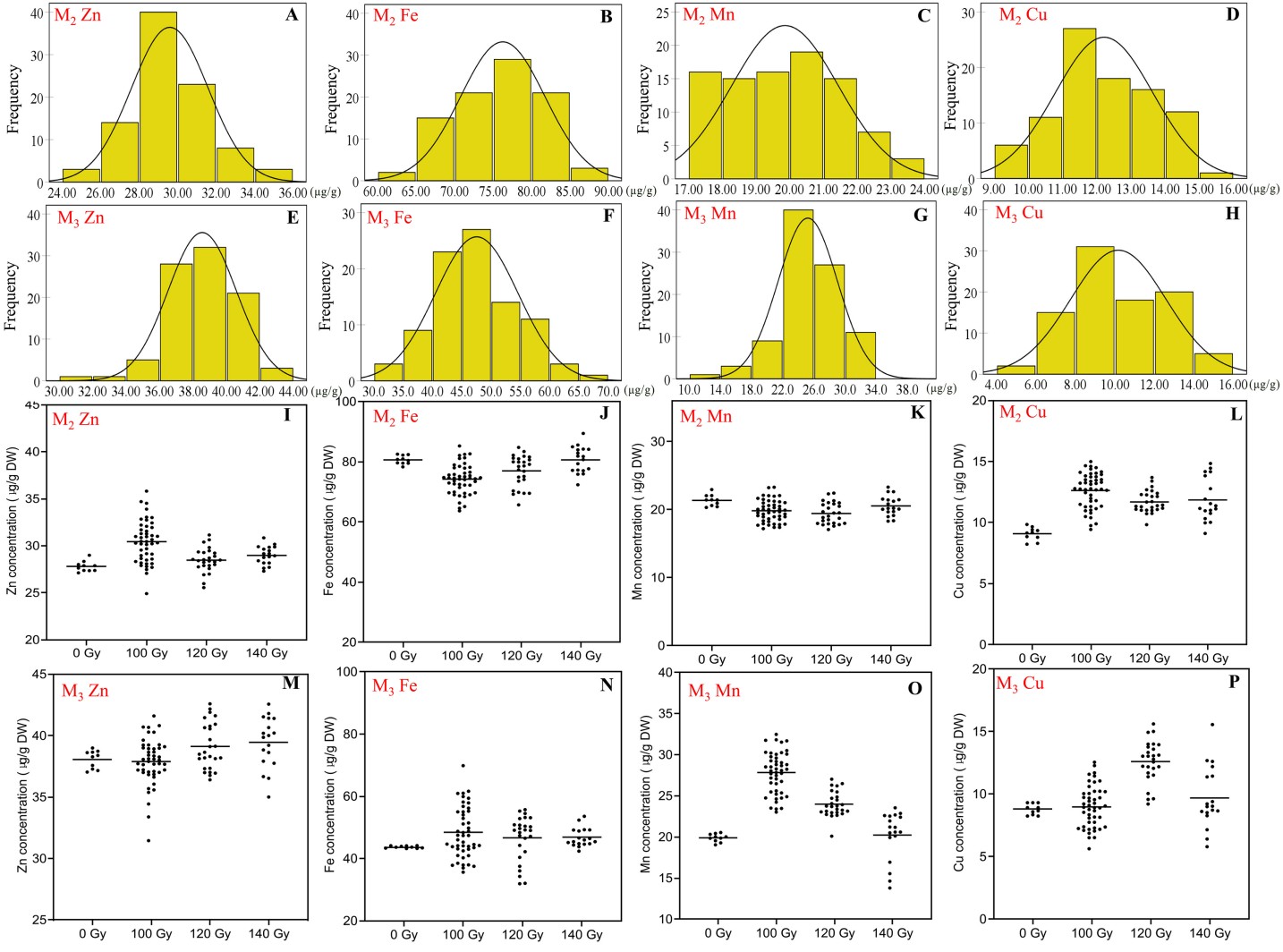

**Figure 3 The distribution of seed trace elements concentration in $M_2$ and $M_3$ generations.** (A and E) Frequency distribution of Zn concentration; (B and F) frequency distribution of Fe concentration; (C and G) frequency distribution of Mn concentration; (D and H) frequency distribution of Cu concentration; (I and M) effect of different irradiation doses on Zn concentration; (J and N) effect of different irradiation doses on Fe concentration; (K and O) effect of different irradiation doses on Mn concentration; (L and P) effect of different irradiation doses on Cu concentration; $M_2$ generation: A–D, I–L; $M_3$ generation: E–H, M–P.           

groups and control in the $M_2$ generation. However, in the $M_3$ generation, the 100 and 120 Gy treatment generally increased Mn concentration compared with control, especially in the 100 Gy group. In the 140 Gy group, the median of Mn concentration was lower than the control group.

The range of Cu concentration in $M_2$ and $M_3$ generations was 9.1–15.0 and 5.6–15.6 μg/g, respectively. In the $M_2$ generation, irradiation treatment generally increased Cu concentration. In the $M_3$ generation, however, Cu concentrations also increased with CIBR treatments, especially in the 120 Gy group.

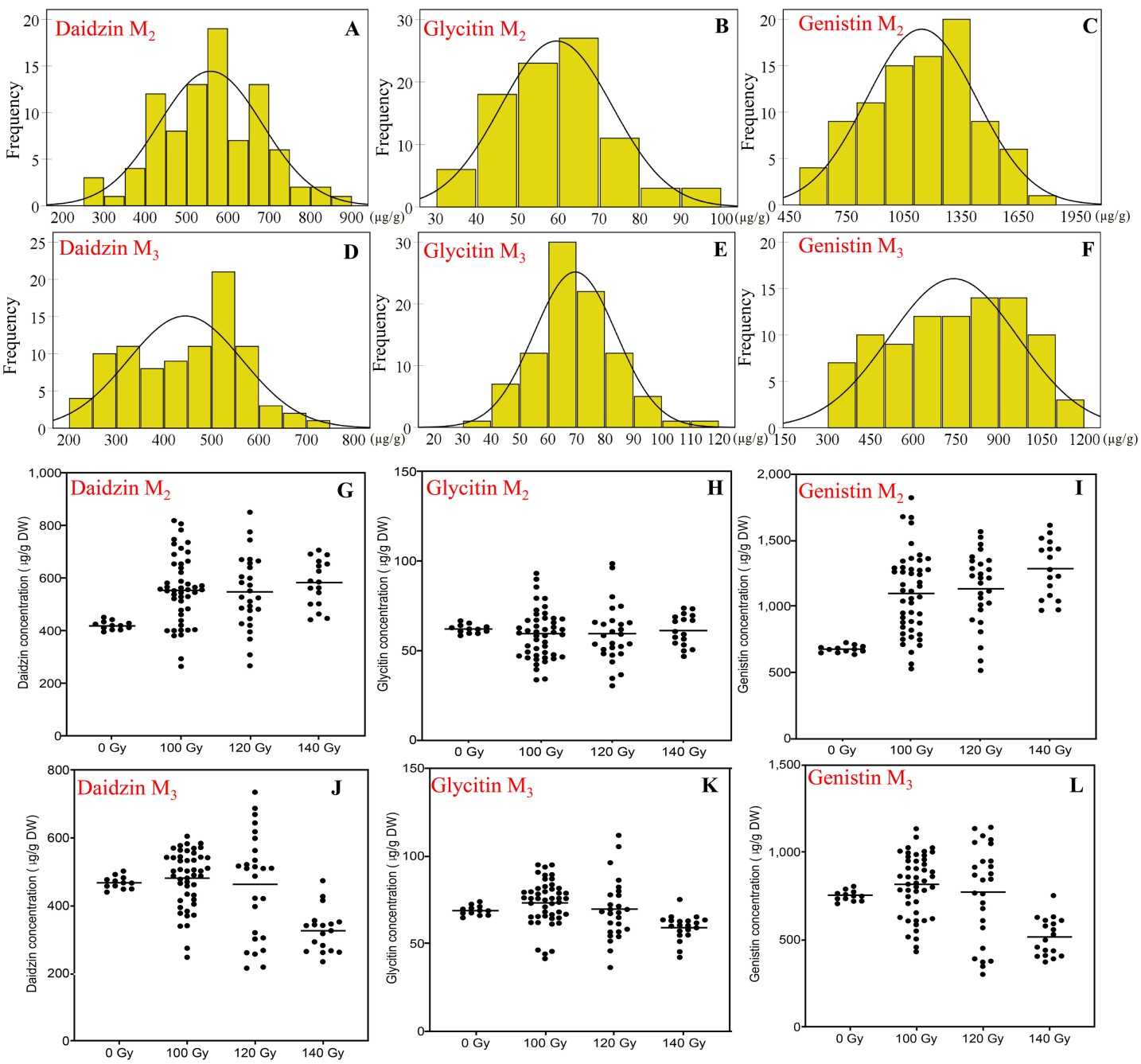

**Figure 4 The distribution of seed isoflavones concentration in M$_2$ and M$_3$ generations.** (A and D) Frequency distribution of daidzin concentration; (B and E) frequency distribution of Glycitin concentration; (C and F) frequency distribution of Genistin concentration; (G and J) effect of different irradiation doses on daidzin concentration; (J and N) effect of different irradiation doses on Glycitin concentration; (K and O) effect of different irradiation doses on Genistin concentration; M$_2$ generation: A–C, G–H; M$_3$ generation: D–F, J–L.

## Effects of CIBR on seed isoflavones concentration

The concentration of different isoflavones components all showed diverse variation in the M$_2$ generation and M$_3$ generation (Fig. 4).

**Table 1 Spearman correlation analyses of seed quality traits in the soybean $M_2$ population.**

| | Protein | Oil | SUG | Sucrose | Zn | Fe | Mn | Cu | Daidzin | Glycitin | Genistin |
|---|---|---|---|---|---|---|---|---|---|---|---|
| Protein | 1.000 | | | | | | | | | | |
| Oil | −0.260* | 1.000 | | | | | | | | | |
| SUG | −0.208* | 0.266* | 1.000 | | | | | | | | |
| Sucrose | 0.127 | 0.030 | −0.144 | 1.000 | | | | | | | |
| Zn | 0.302** | −0.319** | −0.120 | 0.019 | 1.000 | | | | | | |
| Fe | −0.087 | 0.144 | 0.153 | 0.277** | -0.248* | 1.000 | | | | | |
| Mn | −0.030 | 0.017 | −0.132 | −0.076 | −0.004 | 0.075 | 1.000 | | | | |
| Cu | 0.230* | −0.398** | −0.188 | 0.243* | 0.397** | −0.046 | −0.201 | 1.000 | | | |
| Daidzin | −0.026 | 0.090 | −0.057 | 0.020 | −0.133 | −0.061 | 0.043 | 0.075 | 1.000 | | |
| Glycitin | −0.114 | 0.145 | −0.061 | 0.036 | -0.241* | −0.080 | −0.048 | 0.057 | 0.881** | 1.000 | |
| Genistin | −0.046 | 0.181 | 0.072 | 0.137 | −0.048 | 0.071 | 0.120 | −0.050 | 0.715** | 0.649** | 1.000 |

Notes:
* Represent significant difference at $p = 0.05$ level.
** Represent significant difference at $p = 0.01$ level.

To daidzin, in the $M_2$ generation, the range for all CIBR doses was 265–851 µg/g. Compared with control (421 ± 16 µg/g), irradiation treatments generally increased the daidzin concentration. Higher median of the daidzin concentrations was found for all CIBR groups against control, especially in the 140 Gy group. In the $M_3$ generation, the range was 215–735 µg/g. Since the range in control was 468 ± 18 µg/g, the positive effect was decreased. In the group of 120 Gy, about half of lines showed higher daidzin concentrations, while the daidzin concentration of most lines in the group of 140 Gy was lower than control.

Glycitin levels were similar for the two generations, at 30.3–98.6 and 36.5–111.8 µg/g for $M_2$ and $M_3$ generation, respectively. The distribution was different, though there were no significant differences among different irradiation doses in the $M_2$ generation. In the $M_3$ generation, compared with the control, the variation trend of glycine concentration in 100 and 120 Gy groups was similar to that in the $M_2$ generation. While the median of 140 Gy group was lower than that of control.

Genistin had a higher CV compared with the other two isoflavones, the range of genistin concentration reached 514–1,821 µg/g in the $M_2$ generation and 104–1,147 µg/g in the $M_3$ generation. The maximum genistin concentration was 169% and 53% higher than control in the $M_2$ and $M_3$ generation. The treatment of 100 and 120 Gy consistently showed a discrete distribution in the two generations. Therefore, 100 and 120 Gy doses were better for high-genistin mutants selection.

## Correlation analysis of different nutritional quality indexes in $M_2$ and $M_3$ generations

To compare the correlation among different nutritional quality indexes, the correlation of 11 quality indexes was analyzed in this study, as shown in Tables 1 and 2.

In the $M_2$ generation, protein concentration was negatively correlated with oil and soluble sugar, but positively correlated with Zn and Cu concentration. While oil

**Table 2 Spearman correlation analyses of seed quality traits in the soybean $M_3$ population.**

|  | Protein | Oil | SUG | Sucrose | Zn | Fe | Mn | Cu | Daidzin | Glycitin | Genistin |
|---|---|---|---|---|---|---|---|---|---|---|---|
| Protein | 1.000 |  |  |  |  |  |  |  |  |  |  |
| Oil | 0.110 | 1.000 |  |  |  |  |  |  |  |  |  |
| SUG | 0.172 | 0.357** | 1.000 |  |  |  |  |  |  |  |  |
| Sucrose | −0.097 | 0.381** | 0.572** | 1.000 |  |  |  |  |  |  |  |
| Zn | 0.212* | −0.020 | −0.176 | 0.131 | 1.000 |  |  |  |  |  |  |
| Fe | −0.271** | −0.163 | −0.074 | 0.066 | 0.113 | 1.000 |  |  |  |  |  |
| Mn | −0.082 | −0.061 | −0.019 | −0.407** | −0.130 | 0.413** | 1.000 |  |  |  |  |
| Cu | −0.060 | −0.364** | −0.107 | −0.071 | 0.131 | −0.058 | −0.204 | 1.000 |  |  |  |
| Daidzin | −0.101 | 0.002 | 0.135 | −0.105 | −0.281** | 0.216* | 0.383** | −0.132 | 1.000 |  |  |
| Glycitin | −0.153 | 0.035 | 0.088 | −0.062 | −0.200 | 0.117 | 0.268* | −0.164 | 0.756** | 1.000 |  |
| Genistin | −0.084 | 0.007 | 0.107 | −0.137 | -0.339** | 0.232* | 0.448** | −0.119 | 0.961** | 0.655** | 1.000 |

Notes:
* Represent significant difference at $p = 0.05$ level.
** Represent significant difference at $p = 0.01$ level.

concentration was negatively correlated with Zn and Cu concentration. Sucrose concentration was positively correlated with Fe and Cu concentration. Zn concentration was negatively correlated with Fe and glycitin concentration, but positively correlated with Cu concentration. Daidzein, glycitin and genistin concentrations were positively correlated.

In the $M_3$ generation, protein concentration was positively correlated with Zn concentration, but negatively correlated with Fe concentration. Oil concentration was positively correlated with the concentrations of sucrose and soluble sugar, but negatively correlated with Cu concentration. Sucrose and soluble sugar concentrations were positively correlated. Sucrose concentration was negatively correlated with Mn concentration, while Fe and Mn concentrations were both positively correlated with that of daidzin and genistin. Consistent with the $M_2$ generation, daidzein, glycitin and genistin concentrations were positively correlated.

Classifying the 91 lines among the three CIBR doses in $M_3$ generation by hierarchical clustering 11 quality indicators (Fig. 5), four categories: A, B, C and D could be classified as shown in Table 3. Category A involves 16 lines with relatively higher concentration of Zn, Cu, Fe and sucrose; Category B includes 21 lines with relatively higher oil, sucrose and soluble sugar concentration; Category C includes 27 lines with relatively higher protein, Mn and isoflavone concentration; Category D contains 27 lines with higher concentration of soluble sugar, Fe, Mn, Cu and isoflavones.

## DISCUSSION

Soybean seed protein and oil concentration are important quality indicators. Studies have shown that mutation breeding is an effective way to create germplasm resources with high protein and oil concentration (*Chaudhary et al., 2015*). In the present study, the changed range of the protein and oil concentration is about 5%, similar results in fast neutron irradiation methods are also revealed (*Bolon et al., 2014*). Though the negative correlation between protein and oil concentration are still found from the $M_2$ generation to the $M_3$

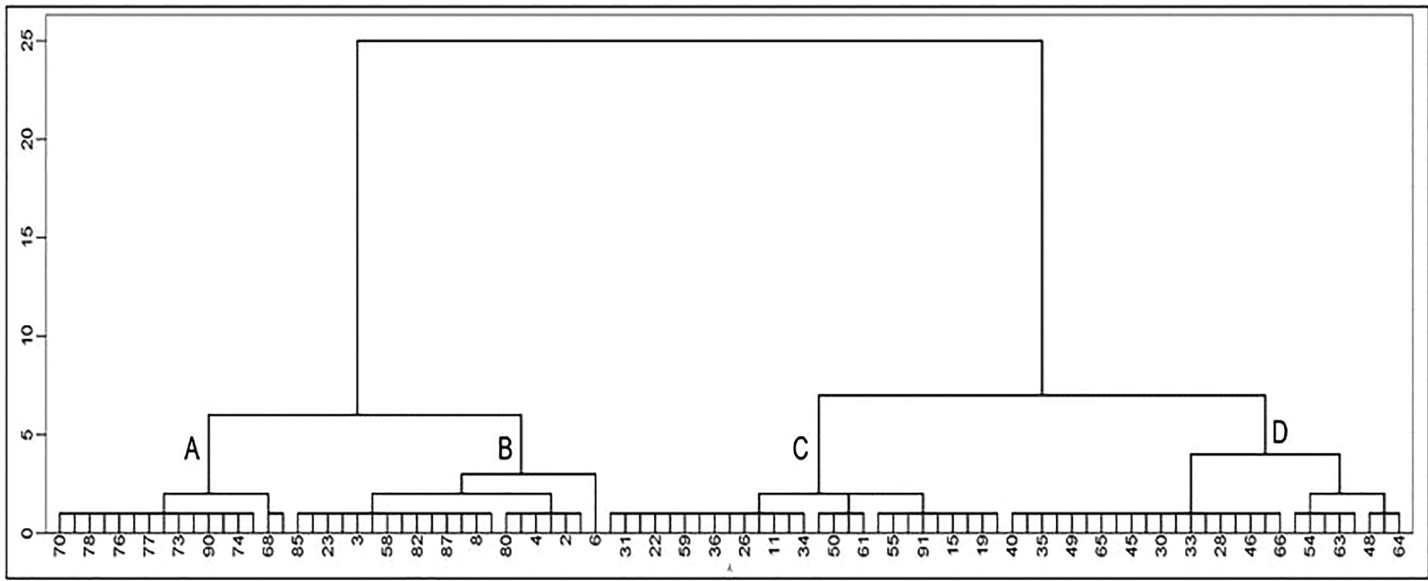

**Figure 5 Cluster analysis of seed quality traits in M₃ generation lines.** (The number in the figure is the line number, 1–47 is the treatment of 100 Gy, 48–73 is the treatment of 120 Gy and 74–91 is the treatment of 140 Gy; the letters A–D represent four categories).

**Table 3 The average of quality traits based on the cluster analysis in M₃ generation.**

| Category | Protein | Oil | SUG | Sucrose | Zn | Fe | Mn | Cu | Daidzin | Glycitin | Genistin |
|---|---|---|---|---|---|---|---|---|---|---|---|
| A | 38.14 | 23.02 | 72.93 | 48.74 | 40.04 | 48.42 | 22.92 | 11.73 | 264.80 | 51.18 | 402.58 |
| B | 38.29 | 24.06 | 77.85 | 49.33 | 38.52 | 44.18 | 23.48 | 9.55 | 368.41 | 65.31 | 584.89 |
| C | 38.75 | 23.35 | 73.91 | 42.29 | 38.28 | 44.90 | 25.72 | 9.51 | 480.40 | 74.57 | 803.09 |
| D | 37.84 | 23.46 | 78.15 | 46.88 | 37.93 | 52.26 | 27.37 | 10.38 | 575.16 | 78.45 | 1,003.17 |

generation, the present study found that within the optimum irradiation dose range, higher irradiation doses induce the high-oil and high-protein mutants with the possibility of stable inheritance. According to the theory of *Patil et al. (2007)*, the irradiation treatment may break the negative correlation between protein and oil. Therefore, higher irradiation dose has the potential in creating lines with both high-oil and high-protein mutants in breeding program.

*Hayashi & Aoki (1985)* once demonstrated that the γ-irradiation treatment could promote the accumulation of sucrose in potatoes. Ethl methane sulfonste (EMS) treatment can increase seed soluble sugar concentration up to two times compared to control in soybean (*Espina et al., 2018*). Consistent with these results, in the present study, a wider range of changes in sucrose and soluble sugar concentrations from the offspring population by carbon ion beam irradiation are also found. The distribution and variation are similar in the M₂ and M₃ generation, and carbon ion beam irradiation treatments generally increase the concentration of soluble sugar and sucrose. There is evidence that irradiation promotes soluble sugar concentration due to restricted breakdown of sucrose and accelerated synthesis, for example, irradiation increases the activity of sucrose phosphate synthase (*Hayashi & Aoki, 1985*). Therefore, the higher irradiation dose within
the optimum irradiation dose range has a more positive effect on sucrose and soluble sugar accumulation, making the trait more easily inherited in early generations.

The content of trace element is another important index to evaluate the nutritional quality of soybean (*Wang, Takematsu & Ambe, 2000*). As a result of radiation exposure, the soybean genes can be mutated, the physiological metabolism of plants can be altered, and mineral elements can be accumulated differently (*Wang et al., 2021b*). The content of plant trace elements is also closely related to the accumulation and metabolism of other nutrient components (*Liu et al., 2019*).

In this study, irradiation treatments have a positive effect on trace element concentrations, but different trace elements respond differently to doses. For instance, the positive effect of 100 Gy dose on Mn element is more apparent than on other trace elements. While, the 120 Gy treatment has the most obvious positive effect on Cu concentration. Trace element distributions in the $M_2$ and $M_3$ generations indicated that these traits are not stabilized in early generations.

Much progress has been achieved in breeding high-isoflavone germplasm resources of soybean through the evaluation of the existing resources and crossing combinations in recent years (*Wu et al., 2020*), it is worthy to mention that Zhejiang University of China screened a high soybean isoflavone mutant (6,100 µg/g) by using the mutagenesis method (*Mei et al., 2014*). In the present study, the concentration of different isoflavones components all shows diverse variation in the two generations. Three main components of isoflavones, daidzin, glycitin, and genistin are detected, among which daizin and genistin are greatly affected by radiation mutagenesis, while glycitin is less affected. Within the optimum irradiation dose range, higher radiation dose (140 Gy) of carbon ion beam reduces the isoflavone concentration in the $M_3$ generation. Since isoflavones are a kind of secondary metabolites, their synthesis pathways are relatively complex with multiple pathways (*Chen et al., 2021*), higher radiation doses might inhibit their synthesis. As a result, higher radiation doses might not be suitable for screening high isoflavone mutants in the early generations. Since environmental factors greatly influence isoflavone concentrations (*Seguin et al., 2004*; *Sivesind & Seguin, 2005*; *Chen et al., 2021*), the application of radiation mutation breeding to screen specific high-isoflavone germplasm still requires more systematic investigation. For example, it might be necessary to conduct repeated experiments with multiple generations and multiple sites for repeated validation.

It is difficult to screen mutants with all excellent nutrition quality indicators in a breeding program, nevertheless, it is feasible to focus on some indicators. Our cluster analysis of the $M_3$ generation lines provides the reference for screening mutants. As the key generations of mutation breeding, the $M_2$ and $M_3$ generations can reflect the mutagenic effects of some traits, however, many nutrition quality traits are still separated in these generations. Therefore, further research is needed to determine in which generation these traits can stably be inherited.

## CONCLUSION

The directional trend induction of soybean quality traits could be realized by adjusting the radiation dose. When the CIBR dosage is within the optimal range, higher doses can

produce both high-oil and high-protein mutants, but not high-isoflavone mutants. However, in the early generations, it is difficult to select trace element traits. In order to accelerate the process of soybean breeding for special purposes, it is essential to determine the optimal dose within the optimal irradiation range aimed at specific breeding target.

### Funding
This work was funded by the National Key R&D Program of China: grant number 2021YFD1201103-03 and the Strategic Priority Research Program of the Chinese Academy of Sciences, Grant Number XDA24030403-3. The funders had no role in study design, data collection and analysis, decision to publish, or preparation of the manuscript.

### Grant Disclosures
The following grant information was disclosed by the authors:
National Key R&D Program of China: 2021YFD1201103-03.
Strategic Priority Research Program of the Chinese Academy of Sciences: XDA24030403-3.

### Competing Interests
The authors declare that they have no competing interests.

### Author Contributions
- Changkai Liu conceived and designed the experiments, performed the experiments, analyzed the data, prepared figures and/or tables, authored or reviewed drafts of the article, and approved the final draft.
- Xue Wang performed the experiments, prepared figures and/or tables, and approved the final draft.
- Yansheng Li analyzed the data, prepared figures and/or tables, and approved the final draft.
- Heng Chen performed the experiments, authored or reviewed drafts of the article, and approved the final draft.
- Qiuying Zhang conceived and designed the experiments, authored or reviewed drafts of the article, and approved the final draft.
- Xiaobing Liu analyzed the data, authored or reviewed drafts of the article, and approved the final draft.

### Data Availability
The raw measurements are available in the Supplemental File.

### Supplemental Information
Supplemental information for this article can be found online at http://dx.doi.org/10.7717/peerj.14080#supplemental-information.

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
