# Peer review of "Irradiation with carbon ion beams affects soybean nutritional quality in early generations"

_PeerJ, doi:10.7717/peerj.14080_

## Round 0.1 · original submission · Major Revisions

Major modification is needed to make it more meaningful. Please revise the manuscript according to the reviewers' comments.

Reviewer 1 ·

Basic reporting

The English throughout the manuscript is clear, unambiguous, and professional.
There are some suggestions:
1. Write the full form of the abbreviated form such as AAS (line 105) and CV (line 165).

Experimental design

1. It would be great if the results of the control experiment is also mentioned along with the other, for example, mention the concentration of protein and oil in the control experiment in the result section along with the M2 and M3 generations experiment. It would help the readers to understand the concept more clearly.

Validity of the findings

1. What is the motive for classifying the lines into four categories. How would it be useful and on what basis do the authors classify them? Mention the technique/approach used for the classification.
2. In the discussion, illustrate the possible reasons behind every result based on the literature. For example, mention the possible reason why carbon ion beam irradiation promotes soluble sugar concentration etc.
3. How did the authors confirm that the change in the concentration of all the nutrients is due to the carbon ion irradiation not due to randomness?

·

Basic reporting

Introduction is written very well with appropriate citations.
But very unfortunately the methodology section is almost incomplete. Some of major the limitations I have note here;
Line 68; Why only one variety? By those 3 treatments, their evaluation and just comparing with control of the same variety (and this was irradiated previously, very confusing!) can never be selected to release as a new variety. Then how are you going to compare your findings with the performance of other varieties those might be irradiated with the same doses of CIBR?
Line 68-69; I am fully confused! In this study what you have done; is it just evaluation of previously irradiated lines? Because the methodology is incomplete even, I couldn't get when and how did you apply the different doses of CIBR on the said only variety. And you are just comparing with one treatment to another treatment and the ultimate effect. To me it can never be enough to make decision to release as new a variety.
Line 72; Brief description is needed; how did you apply the treatment (for general readers).
Line 72; No information about plant growth and development, agronomic practices and so on
Line 73-74; If I am not mistaken M0 means non-irradiated seeds, right? So how those seeds were sown according to the irradiation dose? This one is not clear to me.
Line 76-77; Why there are variations in line numbers? In 100 = 47 lines, in 120 = 26 lines and in 140 = 18 lines; why not similar? Is it depending on numbers of survival plants or what? How did you manage the minimum replications? Clear explanation is necessary.

Line 77; Which methodology did you follow? Please cite.
Line 79; How many seeds were sown for M1 and M2 generations, not reported? How did you know that 100 seeds is enough; please cite the authentic references.

Line 149; What does it mean? In 120 Gy you had only 26 lines, so how have you calculated? 26=100% is it or how? Meaning is that you collected/analyzed each and every lines! Nothing is clear from these results.
Line 196; actually, how many is the total numbers? Have you mentioned it anywhere?

Figure 1-4; Is it percentage (%) or what, need to mention. Applicable for all Figures.

Some other minor issues are highlighted/commented in the reviewed pdf copy (attached).

The results and discussion can be accepted but the methodology must need to change.

So, I recommend “Major Revision”.

Experimental design

No information about number of replications!

Validity of the findings

Detailed reports are given at section 1

Additional comments

Detailed reports are given at section 1

·

Basic reporting

In Abstract and Introduction following comments must be addressed:

Write Iron and Cupper etc at their first mention instead of Fe and Cu

The lines are not easy to comprehend, reframe the sentence “In two generations, the responses of Fe, Mn, Zn, and Cu to irradiation doses were diûerent, but the inhibitory eûect on Mn concentration was enhanced as the irradiation dose was raised”

Experimental design

Authors must address the following comments:

How many seeds were irradiated with each dose of gamma rays?

In which design seeds were planted

How many seeds were sown to raise M2 and M3 generation?

Also mention season of sowing and season of harvesting for each generation.

Add detailed methodology for protein and oil content evaluation

Validity of the findings

The discussion section must be elaborated and the findings of the present study must be put in light of the recent literature. A conclusion section must be added to emphasize the main findings of your present work.

---

## Round 0.2 · Minor Revisions

The English of the manuscript is very poor (see the attached file for some examples, but this is not comprehensive). Before accepting the manuscript for publication the authors should improve the English very carefully.

·

Basic reporting

I have thoroughly checked; the authors have revised the manuscript nicely following comments and suggestions that were made during our first review. They have elaborated the logical explanation under every comments; much appreciated.

Experimental design

Addressed nicely

Validity of the findings

Will contribute significantly to the scientific community

Additional comments

Now the manuscript has reached to an acceptable standard.

---

## Round 0.3 · accepted · Accept

Based on your English polishing, the manuscript can be accepted for publication

·

Basic reporting

They have revised accordingly

Experimental design

no comment

Validity of the findings

no comment